# Zebrafish: A Relevant Genetic Model for Human Primary Immunodeficiency (PID) Disorders?

**DOI:** 10.3390/ijms24076468

**Published:** 2023-03-30

**Authors:** Faiza Basheer, Robert Sertori, Clifford Liongue, Alister C. Ward

**Affiliations:** 1School of Medicine, Deakin University, Geelong, VIC 3216, Australia; 2Institute for Mental and Physical Health and Clinical Translation (IMPACT), Deakin University, Geelong, VIC 3216, Australia

**Keywords:** immunity, immunodeficiency, zebrafish

## Abstract

Primary immunodeficiency (PID) disorders, also commonly referred to as inborn errors of immunity, are a heterogenous group of human genetic diseases characterized by defects in immune cell development and/or function. Since these disorders are generally uncommon and occur on a variable background profile of potential genetic and environmental modifiers, animal models are critical to provide mechanistic insights as well as to create platforms to underpin therapeutic development. This review aims to review the relevance of zebrafish as an alternative genetic model for PIDs. It provides an overview of the conservation of the zebrafish immune system and details specific examples of zebrafish models for a multitude of specific human PIDs across a range of distinct categories, including severe combined immunodeficiency (SCID), combined immunodeficiency (CID), multi-system immunodeficiency, autoinflammatory disorders, neutropenia and defects in leucocyte mobility and respiratory burst. It also describes some of the diverse applications of these models, particularly in the fields of microbiology, immunology, regenerative biology and oncology.

## 1. Introduction

The human immune system is composed of a complex network of specialist cells and organs that represents the body’s major protective mechanism against infections and cancer. Innate immunity, of which neutrophils and macrophages are the key components, represents the first line of defense against these pathogenic threats [1,2]. Adaptive immunity, which is dominated by cells of the lymphoid compartment, provides extended defense, including the immune memory that facilitates rapid and effective responses against subsequent encounters with infectious agents and tumor cells [3]. The development and function of the various immune cells throughout the life-course is precisely controlled by a large set of critical genes.

Human primary immunodeficiencies (PIDs), also known as inborn errors of immunity (IEIs), represent a diverse group of nearly 500 individual disorders [4]. Each is characterized by specific combinations of absent immune cell(s) and/or defective immune function(s) and may include syndromic features outside of the immune system [5]. The major forms typically impact either the lymphoid or myeloid lineages that are the focus of this review (Figure 1). The understanding of how particular genetic defects mediate the specific immunological phenotypes remains poor in many instances. Such information is critical to the development of new and tailored treatment options to overcome the high levels of morbidity and mortality experienced and to provide patient-specific ‘precision’ medical care.

Zebrafish represent an established genetic model for studying development and disease, including the cells and functions relevant to immunity [6]. The zebrafish immune system has both innate and adaptive arms, comprising immune cells that possess comparable functions to their human counterparts, concomitant with a high level of conservation of the specific genes that control their development and action [7]. Zebrafish are also easily manipulable genetically and physically, notably including the introduction of gene mutations using genome editing, as well as transgenesis and other sophisticated approaches [8]. They are also extremely accessible for imaging and other analyses [9] and exhibit favorable husbandry. Collectively, this has resulted in wide application of this organism to study blood and immune cell disorders [10,11], with over 100 publications describing immunodeficient zebrafish, including the gene disruptions associated with human PIDs.

## 2. The Zebrafish Immune System in Comparison to That of Humans

The cells comprising the zebrafish immune system are generated and maintained via distinct waves of hematopoiesis occurring at multiple anatomical sites, as is the case in humans [12]. Primitive macrophages are produced as part of the so-called ‘primitive’ wave of hematopoiesis, some of which populate the brain to form the microglial population, with a transient ‘intermediate’ wave of hematopoiesis in the posterior blood island, generating neutrophils [13]. Multipotent hematopoietic stem cells (HSCs) initiate the ‘definitive’ wave of hematopoiesis and sustain the process throughout the lifespan [14], with these HSCs initially homing to the caudal hematopoietic tissue (CHT), equivalent to the mammalian fetal liver, to produce multiple cell lineages, with lymphocyte precursors generated migrating to the thymus to commence the production of T cells [12,15]. HSCs ultimately populate the kidney marrow, equivalent to the mammalian bone marrow, allowing ongoing production of neutrophils, macrophages and other leukocytes, and later B cells, with full immunity including natural killer (NK) cell populations in place by 4–6 weeks post fertilization [12,15,16].

Like their human counterparts, zebrafish neutrophils expressing the enzymes myeloperoxidase [17] and gelatinase [18] are the initial innate immune cell responders to sites of infection or injury [19], with zebrafish macrophages expressing perforin 2 and l-plastin [20] arriving subsequently to phagocytose pathogens and cell debris [19]. Macrophages and dendritic cells present antigens derived from infecting microorganisms or cancer cells to lymphocytes, initiating an adaptive immune response [21]. Eosinophils and mast cells are also found in zebrafish, with eosinophils able to degranulate in response to antigenic exposure [22,23]. Various zebrafish T cell populations, including those expressing CD4 and CD8, induce appropriate cellular responses mediated via four T cell receptor (TCR) chains (α, β, γ, δ) [15,24], while B cells elicit humoral responses leading to the production of immunoglobin (Ig) molecules equivalent to those seen in mammals (IgM and IgD) as well as fish-specific forms (IgT, IgZ) [25]. Zebrafish also possess two distinct classes of NK cell populations, NK and NK-like, which express archetypal NK-lysin enzymes [26].

The control of immune cell development shows strong correlation between zebrafish and humans. This includes lineage-specific transcription factors important for the production of specific lymphoid and myeloid immune cell lineages. For example, the zebrafish *IKZF1* gene orthologue has been shown to play a key role in early lymphoid development [27], while *SPI1*-related genes were demonstrated to be important in early myeloid cell development [13]. This correlation also extends to cytokine and chemokine cell signaling components that stimulate differentiation, proliferation, survival, migration and activation of these cells [28,29]. As a consequence, zebrafish have been widely employed as a relevant genetic model across multiple categories of PIDs (Table 1).

## 3. Zebrafish Models of Lymphoid Disorders

### 3.1. Severe Combined Immunodeficiency

Severe combined immunodeficiency (SCID) is a disorder that is characterized by a block in T cell differentiation (T−) along with variable impacts on the development of B cells (B+/−) and NK cells (NK+/−), thereby rendering patients highly susceptible to a range of life-threatening infections and cancers [63,64]. 

#### 3.1.1. T−B− SCID

The first zebrafish PID model was generated by targeting the zebrafish orthologue of the human *RAG1* gene. Zebrafish homozygous for a loss-of-function (LOF) *rag1* mutation were viable, but exhibited decreased lymphocytes in both peripheral blood and kidney, with increased circulating neutrophils and increased myelomonocytes in the kidney. The mutants were defective in V(D)J recombination, with rearrangement of TCRβ and IgM significantly abrogated, indicating a paucity of mature T and B cells, although NK markers were found to be comparable to wild-type fish [30,31], and so reflective of the T−B−NK+ SCID observed in patients with *RAG1* deficiency [65]. In contrast, adult zebrafish homozygous for a *rag2* mutation displayed a significant reduction in circulating lymphocytes, reflecting a loss of T cells, but with variable decreases in B cell numbers, possibly due to the hypomorphic nature of the mutation [16,32]. Zebrafish LOF *prkdc* mutants also showed a significant reduction in both T and B cell lineages, with NK cells but not NK-like cells also affected, resulting in high susceptibility to infection and reduced anti-tumor immunity in these fish [16,26,33]. In contrast, LOF mutations and morpholino-mediated knockdown (KD) of the zebrafish *ak2* gene both resulted in reduction of embryonic T cells and neutrophils [34]. These impacts are consistent with the reticular dysgenesis (aleukocytosis) observed in humans deficient in *AK2*, a severe form of T−B−NK− SCID also impacting neutrophils [66].

#### 3.1.2. T−B+ SCID

Human T−B+ SCID is caused by a variety of genetic lesions, with a number of such mutations impacting cytokine signaling via interleukin-2 (IL-2) family members [67], some of which have been successfully modelled in zebrafish. For instance, LOF mutations in the zebrafish *il2rga* gene, which encodes the IL-2 receptor γ (IL-2Rγ) common chain, resulted in severely reduced numbers of embryonic T cells, impaired juvenile TCR rearrangement and significantly reduced adult T cells and NK cells; however, B cells and myeloid cells were not affected [26,35,68]. LOF mutations in *jak3*, encoding the IL-2Rγ-associated Janus kinase 3 (Jak3), shared similar phenotypes, with severely impaired T cell development and a reduction in the number of NK cells [16,36,37], but only minor impacts on B cell maturation and neutrophil development [37]. Both of these T−B+NK− SCID models exhibited perturbed immunity, with the *il2rga* mutants displaying dysregulated intestinal microbiota and defective tumor immunity [35] and the *jak3* mutants being susceptible to the development of lymphoid malignancy [37]. Finally, LOF mutations in *il7r*, encoding a signaling chain utilizing IL-2Rγ, also displayed severely impaired embryonic T cell development [36].

### 3.2. Combined Immunodeficiency

Combined immunodeficiency (CID) disorders are generally less profound than SCID. For example, human ZAP70 (Zeta-chain (TCR)-associated protein kinase, 70 kDa) deficiency represents a rare form of CID, characterized by a loss of peripheral CD8+ T cells and non-functional CD4+ T cells [69]. Consistent with the human disorder, zebrafish *zap70* LOF mutants also displayed severe defects within the T cell lineage, with a reduction in mature T cells in kidney marrow and impaired thymic T cell development observed [38]. Similarly, IKZF1 deficiency impacts memory T and B cell populations in humans [70], with LOF *ikzf1* mutant zebrafish exhibiting a severe reduction in larval T cells, although juvenile and adult T cell development was less impacted [39].

### 3.3. Multi-System Immunodeficiency

Other human immunodeficiencies have concurrent associated or syndromic features and so are referred to as ‘multi-system’ disorders [4]. For example, growth hormone insensitivity syndrome with immune dysregulation 1 (GHISID1) is characterized by immune perturbation and postnatal growth failure due to LOF mutations in human *STAT5B* [71]. LOF mutations in the equivalent zebrafish gene, *stat5.1*, caused impaired lymphopoiesis with reduced T cells throughout the lifespan, along with broader disruption of the lymphoid compartment into adulthood, including evidence of increased T cell activation [40]. The mutants also showed reduced growth and increased adiposity with concomitant dysregulation of growth and lipid metabolism genes [40,72], faithfully recapitulating the human disease. Similarly, human hypomorphic *EXTL3* mutations cause an immuno-osseus syndrome that involves both immune deficiency and skeletal dysplasia [1], with equivalent zebrafish LOF *extl3* mutants showing decreased thymic T cells [41] and defective pectoral fin development [73]. In addition, knockdown of the zebrafish *arpc1b* gene resulted in reduced embryonic T cells and thrombocytes [42], recapitulating human *ARPC1B* deficiency [74]. Ataxia-telangiectasia (A-T) syndrome is a human disorder characterized by neurodegeneration, immune dysregulation, cancer susceptibility and premature aging and results from defects in the *ATM* gene [43]. Zebrafish LOF *atm* mutants exhibited similar phenotypes, with multi-lineage immunodeficiency along with motor disturbances and cancer predisposition [75]. Finally, human *BCL11B* gain-of-function (GOF) mutations have been identified in an autosomal-dominant syndrome characterized by reduced T cells, congenital craniofacial abnormalities and neurocognitive defects [76], with overexpression of a patient-derived BCL11B mutant in *bcl11b* KD zebrafish embryos able to model the T cell defects and dominant nature of the human disease [44]. Other mutations impact human thymus development and thereby cause immunodeficiency indirectly by abrogating T cell production at this site [77]. This has been recapitulated in zebrafish *foxn1* LOF mutants [78], both *chd7* LOF mutants and KD embryos [46], and zebrafish *tbx1* LOF mutants [47].

### 3.4. Auto-Inflammatory Disorders

A variety of human PIDs result from enhanced inflammation, including several auto-inflammatory disorders and immunodeficiencies [4]. For example, *NCKAP1L* deficiency has been shown to result in immunodeficiency, lymphoproliferation and excessive inflammation, with zebrafish *nckap1l* KD shown to cause defective neutrophil migration [48].

## 4. Zebrafish Models of Myeloid Disorders

### 4.1. Congenital Neutropenia

A number of human PIDs are characterized by insufficient numbers of neutrophils, with several of these successfully modelled in zebrafish. For example, LOF mutations in human *CSF3R*, encoding the receptor for the cytokine granulocyte colony-stimulating factor (G-CSF), have been identified in a cohort of neutropenia patients unresponsive to G-CSF treatment [79]. Zebrafish *csf3r* LOF mutants also possess significantly decreased numbers of neutrophils from embryonic to adult stages [49,80]. The neutrophils were found to be functionally compromised, with mutants unable to respond to G-CSF and displaying enhanced susceptibility to bacterial infection [49]. LOF mutations in *SMARCD2*, encoding a chromatin remodeling factor, have also been associated with human neutropenia in combination with myelodysplasia and developmental abnormalities, with ablation of the zebrafish *smarcd2* gene similarly leading to a reduction in neutrophil numbers in embryos [50]. Mutations in human Wiskott-Aldrich syndrome (WAS) protein have been implicated in the pathogenesis of X-linked neutropenia (XLN) [74]. Consistent with this, a LOF zebrafish *was* mutant showed defective migration of neutrophils as well as macrophages, which correlated with increased susceptibility to bacterial infection due to delayed pathogen clearance [51]. Similarly, human *HAX1* mutations have been associated with autosomal-recessive severe congenital neutropenia (SCN) [81], with *hax1* KD in zebrafish embryos resulting in impaired neutrophil development [52]. In addition, LOF mutations in human *VPS45*, encoding a regulator of endosomal membrane trafficking, have been associated with neutropenia, neutrophil dysfunction, nephromegaly and bone marrow fibrosis, with the reduced neutrophil count recapitulated in zebrafish *vps45* KD embryos [53]. Finally, patients with Shwachman-Diamond syndrome present with neutropenia in combination with exocrine pancreas deficiency and skeletal muscle abnormalities associated with dominant mutations in the ribosome maturation factor SRP54 in some cases [82]. Equivalent zebrafish *srp54* mutants exhibited decreased embryonic neutrophils, although this was only observed in homozygous animals, indicating a lack of dominance in zebrafish [54].

### 4.2. Motility Defects

Mutations in the human *ITGB2* gene encoding the cell surface molecule CD18 have been associated with human leukocyte adhesion deficiency (LAD), a disorder characterized by recurrent infections and reduced neutrophil motility [83]. Consistent with this, zebrafish LOF *itgb2* mutants showed decreased trafficking to sites of inflammation despite increased neutrophil numbers in circulation [55]. Similarly, mutations in the *RAC2* gene, encoding a member of the small GTPase family, have been associated with immunological defects in humans, with dominant-negative mutations leading to granulocyte mobility defects [84]. Ablation of the zebrafish *rac2* gene by mutation or KD resulted in reduced mobility of neutrophils and macrophages, but enhanced neutrophil mobilization, resulting in hypersensitivity to infection by *Pseudomonas aeruginosa* [56,57]. So-called ‘warts, hypogammaglobulinemia, infections, and myelokathexis’ (WHIM) syndrome is a PID disorder characterized by defective and decreased neutrophils and consequent recurrent infections, mediated by dominant mutations in the human *CXCR4* gene that encodes a G-protein-coupled chemokine receptor [85]. Zebrafish embryos transgenically expressing the equivalent mutations in the context of the zebrafish CXCR4 equivalent, Cxcr4b, possessed reduced numbers of embryonic neutrophils with impaired neutrophil mobility and wounding recruitment, along with enhanced retention in the hematopoietic tissues [58], reminiscent of the human WHIM disorder.

### 4.3. Respiratory Burst Defects

The neutrophil respiratory burst is a key mechanism for mediating the killing of bacteria and other pathogens but also plays an important role in cell signaling. Human chronic granulomatous disease (CGD) is caused by mutations in a number of genes that encode components of the so-called ‘phagocyte NADPH oxidase’ (PHOX) complex, which mediates superoxide formation, including the *CYBA* gene encoding the p22Phox subunit [86]. Zebrafish LOF *cyba* mutants showed reduced neutrophil reverse migration and impaired macrophage wound attraction, leading to increased neutrophil infiltration, enhanced susceptibility to invasive fungal infection and neutrophil-mediated inflammation [59,60]. The myeloperoxidase (MPO) enzyme acts on H_2_O_2_ generated in injured tissues through the respiratory burst, with evidence that humans harboring *MPO* mutations show increased susceptibility to infections [87]. Zebrafish *mpo* LOF mutants had normal numbers of neutrophils that migrated faithfully to sites of infection [61,62], with similar survival following infection with the fungal pathogen *Candida albicans*; however, there was increased pathogen proliferation, neutrophil accumulation and elevated expression of inflammatory cytokines [62].

## 5. Other Zebrafish Immunodeficiency Mutants

Additional zebrafish immunodeficient models have been generated that resemble human PIDs. For example, the zebrafish *earl gray* mutant, which represents a LOF mutation in the gene encoding the p110/SART3 general splicing factor, exhibits disrupted thymic development, leading to thymic hypoplasia and a lack of T lymphocytes typical in forms of CID [88]. In addition, LOF *runx1* mutations resulted in a reduction in B cells and a failure in B cell V(D)J rearrangement, a phenotype similar to that observed in human common variable immunodeficiency [89]. Furthermore, zebrafish *tlx1* mutations were found to disrupt spleen development, resulting in partial impairment of mononuclear phagocytes and reduced levels of IgM [90]. Other mutants have identified novel regulators of immune cell development and/or function, with the underlying genes representing potential candidates for idiotypic human PIDs. For example, a missense mutation in the transcriptional regulator *zbtb17* resulted in an early block in zebrafish intrathymic T cell development [91], while a *cmyb* mutant zebrafish displayed severely reduced lymphocytes, precursors and myelomonocytes in the kidney, with early lethality [92]. In addition, the zebrafish *urb2* mutant was deficient in HSC within the CHT and early T cells in the thymus [93], whereas *stn1* KD zebrafish displayed an arrest in T cell progenitors [94]. Zebrafish is also being used to identify immune defects as part of other complex human syndromic disorders. For example, a fatal syndrome characterized with severe autoinflammation and leukoencephalopathy due to glycogen-storage-associated mitochondriopathy was identified in eight sporadic families with *C2ORF69* LOF mutations, with the immune perturbation verified in *c2orf69* mutant zebrafish [95]. Zebrafish are additionally being employed to confirm the gene–phenotype correlation in emerging human PIDs, such as LOF defects in human *SRP19* and *SRPRA* associated with SCN [96]. Finally, this model organism is being applied to the study of other key aspects of immunity, such as barrier immunity [97] and Toll-like receptors (TLRs) [98].

## 6. Applications of Zebrafish Immunodeficiency Models

Immunodeficient zebrafish lines, including PID models, are being widely employed to offer new insights into the interplay of the immune system with pathogenic agents and normal microbiota, as well as in the fields of regenerative medicine and cancer research (Figure 2).

### 6.1. Infection and Host Responses

Immunodeficient zebrafish models have served as excellent in vivo platforms for studying host–pathogen interactions involving a wide variety of bacteria and other microorganisms. In many cases these studies take advantage of the optical transparency of zebrafish, often combined with fluorescent labeling of specific immune cell lineages and/or microbial agents to provide unprecedented levels of detail [99]. Utilization of various immune cell depletion strategies has allowed the role of specific lineages to be definitively determined [19]. For example, zebrafish *rag1* mutants were found to be highly susceptible to *Mycobacterium marinum* infection, demonstrating a heavy reliance on adaptive immunity to defend against these pathogens [100]. However, reinfection with the same bacteria highlighted the ability of innate immune cells to mediate an ‘adaptive’ response, with vaccinated mutants displaying increased survival compared to naïve mutants [101]. Enhanced susceptibility to bacterial infection was demonstrated in both *csf3r* [49] and *myd88* [102] LOF mutants, highlighting the essential functions of innate immunity in defense against bacteria. Increasingly, these studies are being directed toward human pathogens [103]. Thus, analysis of the adult T cell–deficient zebrafish *lck* mutant strain has revealed the significant role played by T cell–mediated immune response in immunity against the clinically relevant pathogen *M. abscessus* [104]. Other studies have investigated the role of specific elements of immunity in infection, such as the function of neutrophil calprotectin in immunity to *Vibrio cholerae* [105].

### 6.2. Immunity and Gut Microbiome

Immunodeficient zebrafish have also been used as important tools to study the immune system under normal and diseased conditions. This has provided new insights into immune development, but also inflammation and its resolution [106], allowing dissection of the signaling pathways that regulate recruitment and fate of phagocytes at inflammatory sites [107]. Crossing of PID models onto the optically transparent *casper* background has further enhanced the ability to characterize immune cells, allowing sophisticated approaches to be employed, including transcriptome analysis via RNAseq [16,26]. More recently, PID and other zebrafish models have been applied to study the interplay between immunity and the gut microbiome, typically analyzed by so-called ‘Next-gen’ sequencing of genomic DNA extracted from dissected gut tissues, complemented by the ability to manipulate the microbiome and also apply cutting-edge imaging approaches [108,109]. For example, zebrafish *il2rga* mutants displayed a dysregulated intestinal microbiota with significantly reduced alpha diversity and a concomitant increase in short chain fatty acid (SCFA)-producing bacteria [35], similar to human SCID patients [110]. In contrast, LOF *rag1* mutants showed altered abundance of *Vibrio* species in the gut [111]. The contribution of innate immune cells to microbiota composition was revealed using LOF *myd88* [112] and *irf8* [113] mutants, both of which showed significantly altered gut microbiome composition.

### 6.3. Tissue Repair and Regeneration

The immune system plays a crucial role in tissue repair and regeneration. The regenerative mechanisms of a range of tissues, such as spinal cord, heart, retina, liver, pancreas, nervous system and skeletal elements, have been explored using zebrafish models, taking advantage of the ease of manipulation and imaging in this organism [114,115,116]. In particular, the use of zebrafish deficient in specific subsets of immune cells has aided in determining their functional roles in key aspects of regenerative biology. For example, the use of *foxp3a* mutant zebrafish has revealed an essential function for Treg cells in producing regenerative factors tailored to specific tissue repair/regeneration [117]. Zebrafish neutrophils and macrophages have also been shown to be critical in tissue repair and regeneration, which extends to microglial cells in the brain [118,119]. For example, *granulin*-deficient zebrafish lacking embryonic macrophages and embryonic and adult neutrophils showed impaired tissue repair and wound healing [120], whereas *cftr* deficiency resulted in excessive neutrophil recruitment, leading to increased tissue damage and abnormal repair [121]. In contrast, analysis of the neutrophil-defective *runx1* mutant has revealed an unexpected inhibitory role for neutrophils during regeneration of the tail fin [118]. 

### 6.4. Cancer Xenotransplantation

Amongst the diverse applications of zebrafish immunodeficiency models, their use in cancer xenotransplantation studies continues to show great promise, from a handful of papers prior to 2009 to now over 200 publications. These are adding to the understanding of cancer biology and aiding the development of therapeutics [120,122]. A wide range of human cancer cell lines and tumors have been successfully grafted in zebrafish and analyzed using a variety of imaging-based strategies to investigate tumor cell proliferation, apoptosis, invasion, metastasis, angiogenesis and interactions with the host, including the tumor microenvironment, or for pre-clinical testing of potential therapeutic compounds by placing them directly in the water [123,124,125,126,127]. For example, the *rag2* mutant line has been shown to be a useful model for human cancer xenotransplantation [32], while zebrafish *prkdc* mutants show 50–70% engraftment when injected intraperitoneally with tumor cells from human melanoma, leukemia, pancreatic cancer and bile duct cancer cell lines [33]. Comparison of zebrafish *prkdc*, *rag2* and *jak3* mutant models has revealed the *prkdc* mutants as the most efficient platform for tumor xenotransplantation [128]. Crossing of PID models onto the *casper* background has further enhanced the ability to image and characterize fluorescently labeled cancer cells [37] as well as immune cells and the interactions between the two in real time [129]. For example, a *prkdc casper* mutant line was used to develop a platform for drug testing of fluorescently labelled huma leukemia xenografts [130]. A *prkdc il2rga casper* mutant line was demonstrated to successfully engraft a wide range of human cancer cells that could be monitored in vivo at single-cell resolution [131]. Finally, a *rag2 il2rga casper* mutant line of zebrafish was suitable for long-term engraftment of human cancer and T cells, allowing it to be used as an anticancer screening platform to quantify responses to a range of cancer immunotherapy approaches, including those involving chimeric antigen receptor (CAR) T cell, bispecific T cell engagers (BiTE), and antigen-peptide epitope conjugate (APEC) approaches [132]. In addition to human cancer cell line engraftment, zebrafish SCID models have been successfully applied to the engraftment of primary patient-derived samples, such as patient-derived xenografts (PDXs) or patient-derived organoids (PDOs) [131,133,134]. These have been utilized for the assessment of tumor heterogeneity [135], chemosensitivity [136], radiosensitivity [137] and high-throughput therapeutic drug and immunotherapy testing [138]. 

### 6.5. Other Applications

Zebrafish immunodeficiency models have also been employed for additional applications. For example, zebrafish are being used to assess the relative impact of different human mutations, such as by transgenic expression of WAS protein mutants in a *was* knockout model [51]. Additionally, the zebrafish models are allowing for increased mechanistic understanding. This includes broader pathways involved in mediating the observed immunodeficiency. For example, both JAK1 and JAK3 have been implicated in IL-7R-mediated SCID [36], and IL-2R and JAK1 in JAK3-mediated SCID [37], with downstream STAT5 playing a role in the SCID phenotypes mediated by either IL-2Rγ [35] or JAK3 [37]. Similarly, analysis of *hax1* KD zebrafish has identified a key role for decreased expression of target genes within the G-CSF pathway, which could be reversed by the addition of G-CSF [52]. This approach has been extended to understanding the complex genetic interactions across a suite of genes that cause T cell deficiency when mutated in zebrafish [139]. Zebrafish are further contributing to new biochemical understanding of these diseases, such as decreased lipid storage and ribosomal proteins within neutrophils following *srp54* ablation [54], as well as an abnormally sustained H_2_O_2_ burst in *mpo* mutants, indicating a crucial role in H_2_O_2_ downregulation [61,62]. Such biochemical knowledge is leading to the development of new strategies for therapy. For example, localized activation of a photoactivable Rac-GTP has been identified as a novel therapeutic approach to overcome the defective neutrophil trafficking due to constitutive CXCR4/SDF-1 signaling [58].

## 7. Limitations and Future Prospects

Collectively the studies presented illustrate the strengths of zebrafish as a model for human PIDs. However, there are also a number of limitations that need to be considered. For example, many infection studies have concentrated on embryonic and larval stages, prior to full adaptive immunity being present, providing an incomplete picture of host responses [107]. Microbiome studies also need to be interpreted with some caution due to differences in intestinal anatomy and core microbiota [140], including the absence of strictly anaerobic bacteria in zebrafish [109], with longitudinal studies not possible [141]. Regeneration studies have been generally limited to zebrafish tissues [115,116], making the translation to human cells and tissues difficult. Finally, xenotransplantation studies typically employ compromised temperatures and other conditions to enable human cells to be propagated in zebrafish [120]. For some studies, mouse models may be superior, although they too have limitations [142].

However, the full prospects for zebrafish as a model remain untapped. Future infection studies have the potential to fully investigate host determinants of disease and, combined with chemical screens, to identify potential in patient-specific ‘precision’ therapeutics [107]. Microbiota studies can be extended to simultaneously manipulate the host, bacteria and environment to obtain unprecedented insights into the interplay between these [141]. Regeneration studies can also consider other immune populations, including for example, adaptive immune cells, dendritic cells, mast cells and M1 versus M2 macrophages [19,106]. Future cancer studies can further study the role of angiogenesis [122] and better mimic the human tumor microenvironment [129], including creating humanized models as achieved in mice [142]. Finally, so-called cancer ‘avatars’ have application in patient-specific ‘precision’ cancer therapy pipelines, with the potential to directly predict patient outcomes and thus influence individual patient care [143,144].

## 8. Conclusions

Zebrafish shows a strong correlation of immune cells and the genes controlling their development and function with humans. This, in combination with the genetic malleability and accessibility of this organism, has underpinned its broad application to modelling of a wide range of human PIDs, including multiple examples of both T−B− and T−B+ SCID, CID, multi-system ID, autoinflammatory disorders, congenital neutropenia as well as leucocyte mobility and respiratory burst defects. This has allowed precise and robust analysis of these genetic disorders without the limitations of human studies, providing exciting new insights into these diseases. In addition, a variety of zebrafish immunodeficiency models are being applied to the study of infection, immunity, the microbiome, tissue regeneration and repair, as well as tumor biology. This is leading to unprecedented new knowledge and novel possibilities for treatment. This includes true precision medicine, where an individual patient’s specific mutation(s) can be modeled—and various tailored therapies extensively tested—employing zebrafish models. There are limitations to the use of zebrafish but also unique opportunities unavailable in other experimental platforms. Collectively, this shows that zebrafish is indeed a relevant genetic model for human PIDs.

## Figures and Tables

**Figure 1 ijms-24-06468-f001:**
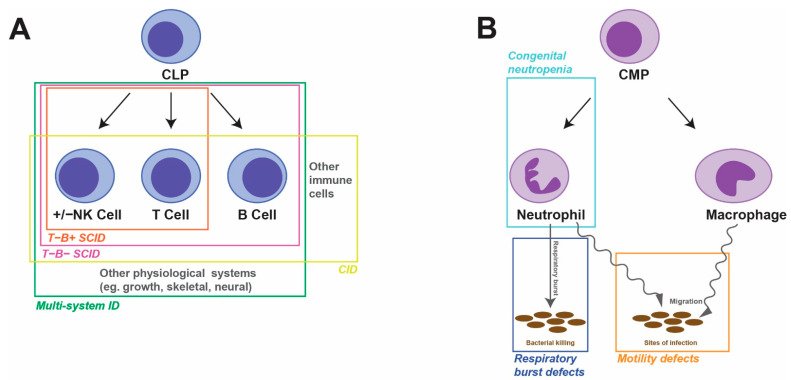
Overview of human PIDs. Schematic of lymphopoiesis (**A**) and myelopoiesis (**B**), indicating the cells, developmental pathways and activities impacted by the indicated PIDs.

**Figure 2 ijms-24-06468-f002:**
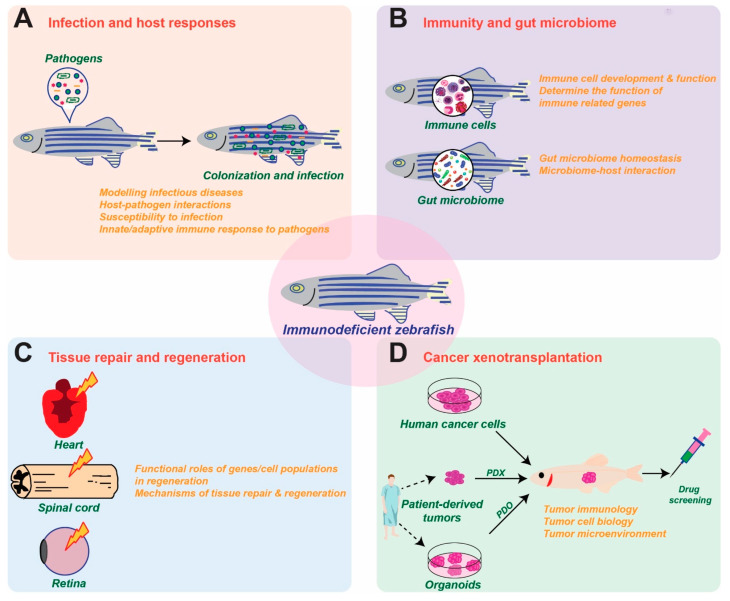
Applications of immunodeficient zebrafish models. Schematic diagrams of how immunodeficient zebrafish are being utilized to study infection and host responses (**A**), immunity and the gut microbiome (**B**), tissue repair and regeneration (**C**) and cancer xenotransplantation (**D**).

**Table 1 ijms-24-06468-t001:** Zebrafish models of human PIDs.

Human Defect ^†^	Inheritance	OMIM	Zebrafish Model *	Immune Phenotype #	References
**Lymphoid deficiencies**					
**T−B− severe combined immunodeficiency** (**SCID**)					
*RAG1*	AR	179615	*rag1* LOF	↓ T and B cells	[30,31]
*RAG2*	AR	179616	*rag2* hypomorph	↓ T cells, variable B cell deficiency	[16,32]
*PRKDC*	AR	615966	*prkdc* LOF	↓ T, B and NK (but not NK-like) cells	[33]
*AK2*	AR	103020	*ak2* LOF + KD	↓ embryonic T cells (and neutrophils)	[34]
**T−B+ severe combined immunodeficiency** (**SCID**)					
*IL2RG*	XL	308380	*il2rga* LOF	↓ T and NK cells	[26,35]
*JAK3*	AR	600173	*jak3* LOF	↓ T and NK cells, minor defects in mature B cells and neutrophils	[16,36,37]
*IL7R*	AR	146661	*il7r* LOF	↓ embryonic T cells	[36]
**Combined immunodeficiencies** (**CID**)					
*ZAP70*	AR	269840	*zap70* LOF	↓ thymic and mature kidney T cells	[38]
*IKZF1*	AD	603023	*ikzf1* LOF	↓ embryonic T cells, less severe defect in juveniles and adults	[39]
**Multi-system immunodeficiencies**					
*STAT5B*	AR	245590	*stat5.1* LOF	↓ T cells throughout lifespan, ↑ T cell activation	[40]
*EXTL3*	AR	617425	*extl3* LOF	↓ embryonic T cells	[41]
*ARPC1B*	AR	604223	*arpc1b* KD	↓ embryonic T cells	[42]
*ATM*	AR	607585	*atm* LOF	↓ embryonic T cell and neutrophils, ↓ lymphocytes and ↑ precursors in adults	[43]
*BCL11B*	AD	617237	*BCL11B* GOF TG	↓ embryonic T cells	[44]
*FOXN1*	AR	601705	*foxn1* LOF	↓ embryonic T cells	[45]
*CHD7*	AD	608892	*chd7* LOF + KD	↓ embryonic T cells	[46]
*TBX1*	AD	602054	*tbx1* LOF	athymic	[47]
**Autoinflammatory disorders**					
*NCKAP1L*	AR	618982	*nckap1l* LOF	defective neutrophil migration	[48]
**Myeloid deficiencies**					
**Congenital neutropenias**					
*CSF3R*	AR	138971	*csf3r* LOF	↓ neutrophils throughout lifespan	[49]
*SMARCD2*	AR	601736	*smarcd2* LOF + KD	↓ embryonic neutrophils	[50]
*WAS*	XL	300392	*was* LOF	defective neutrophil and macrophage migration	[51]
*HAX1*	AR	605998	*hax1* KD	↓ embryonic neutrophils	[52]
*VPS45*	AR	615288	*vps45* KD	↓ embryonic neutrophils	[53]
*SRP54*	AD	604857	*srp45* LOF	↓ embryonic neutrophils	[54]
**Motility defects**					
*ITGB2*	AR	600065	*itgb2* LOF	defective neutrophil trafficking to inflammatory sites	[55]
*RAC2*	AD	608203	*rac2* LOF + KD	defective neutrophil and macrophage mobility	[56,57]
*CXCR4*	AD	162643	*cxcr4* GOF TG	↓ embryonic neutrophils, defective neutrophil mobility and wound recruitment	[58]
**Respiratory burst defects**					
*CYBA*	AR	608508	*cyba* LOF	defective neutrophil reverse migration and macrophage wound attraction	[59,60]
*MPO*	AR	254600	*mpo* LOF	sustained hydrogen peroxidase burst	[61,62]

^†^ Limited to those listed in the 2022 update of human inborn errors of immunity by the IUISEC [4]. * GOF: gain-of-function; KD: knockdown; LOF: loss-of-function; TG: transgenic. # ↓: decreased; ↑: increased.

## Data Availability

No new data was created or analyzed in this study.

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
