# Peer review of "Zebrafish: A Relevant Genetic Model for Human Primary Immunodeficiency (PID) Disorders?"

_ijms, 2023, doi:10.3390/ijms24076468_

Round 1

Reviewer 1 Report

This Review is an extensive overview on the potential application of Zebrafish to pathogenetic studies of IEIs. It is well written and to my knowledge complete in all the paragraphs. I would only include the applicability to intrinsic defects of thymus, as Pax or Foxn1 defects

Author Response

This Review is an extensive overview on the potential application of Zebrafish to pathogenetic studies of IEIs. It is well written and to my knowledge complete in all the paragraphs. I would only include the applicability to intrinsic defects of thymus, as Pax or Foxn1 defects

  • We thank this reviewer for their comments. It is a little unclear what they are specifically requesting, but we have clarified that the focus is on those human PIDs included in the most recent update from the relevant international committee [Tangye et al. 2022].

Reviewer 2 Report

The manuscript by Ferguson et al., submitted in the journal IJMS, entitled “Zebrafish as a Genetic Model for Primary Immunodeficiency (PID) Disorders", This review aims to present the zebrafish immune system, detailing the use of zebrafish as a genetic model for a variety of human IDPs, and describes some of the many applications of these models.

Points that need to be reviewed about the manuscript

I consider that an animal model option is the use of zebrafish, but it has several disadvantages compared to other animal models, in this sense, the disadvantages could also be mentioned, for example, it is restricted to longitudinal studies, among others.

The subjects addressed in the manuscript were adequately structured, but what is needed is to show technical-scientific information on the models and this could be done by creating some tables that reflect, for example, each area indicated in figure 1. In these tables, details of some studies, such as evaluation techniques, evaluation times, analysis tools, among other data. This information will be of great importance to the reader.

A point that could be put in numerical form is the growth in the use of this model over time.

It could be informed what are the limitations of the study.

I believe that this manuscript brings relevant information, but that the information mentioned above needs to be complemented.

Author Response

The manuscript by Ferguson et al., submitted in the journal IJMS, entitled “Zebrafish as a Genetic Model for Primary Immunodeficiency (PID) Disorders", This review aims to present the zebrafish immune system, detailing the use of zebrafish as a genetic model for a variety of human IDPs, and describes some of the many applications of these models.

 Points that need to be reviewed about the manuscript

I consider that an animal model option is the use of zebrafish, but it has several disadvantages compared to other animal models, in this sense, the disadvantages could also be mentioned, for example, it is restricted to longitudinal studies, among others.

  • As suggested, discussion of disadvantages of zebrafish as an animal model has been included in a new section entitled ‘Limitations and Future Prospects’.

The subjects addressed in the manuscript were adequately structured, but what is needed is to show technical-scientific information on the models and this could be done by creating some tables that reflect, for example, each area indicated in figure 1. In these tables, details of some studies, such as evaluation techniques, evaluation times, analysis tools, among other data. This information will be of great importance to the reader.

  • As suggested, additional details of techniques and tools used in evaluating the various models has been included. This has been added to relevant sub-sections of Section 6 rather than additional tables to provided better context.

A point that could be put in numerical form is the growth in the use of this model over time.

  • As suggested some numerical indication of the growth of this model has been included in relevant parts of Sections 1 and 6.

It could be informed what are the limitations of the study.

  • In response to this suggestion, the scope of the study is restricted to those human PIDs to those included in the most recent update from the relevant international committee [Tangye et al. 2022]. Limitations of zebrafish models are incorporated in the new ‘Limitations and Future Prospects’ section.

I believe that this manuscript brings relevant information, but that the information mentioned above needs to be complemented.

  • Noted.

Reviewer 3 Report

Review report -ijms-2294851

Zebrafish as a genetic model for primary immunodeficiency (PID) disorders

Title: The present title seems very rudimentary so needs to put some more effort to revise the title and it should end with question mark. Change the title to make it more catch

Abstract:

In abstract section a flow is missing so needs to be rephrased. Need to mention the significance of the Zebra fish. This part is lacking of objective and scope of the MS. What it will add to existing knowledge.

Introduction

 I suggest authors to put 80 % references should be beyond 2018. The old references can be replaced with latest references. Author are suggested to following reference as standard reference.

 DK Meena, BK Behera, Das Pronob, AK Prusty, Kumar Satendra, M Sekar, Meena Kanti (2012). Transposable elements: strategies and mechanism of transposition in Danio rerio, a genetic model. Asian Journal of Bio Science.7(2):223-229.

Need to extend the introduction part.

2. The zebrafish immune system, this heading can be changed as, The zebrafish immune system v/s immune system of higher vertebrates.

3. Zebrafish models of lymphoid disorders, this part can be represented a schematic diagramme of various.

4. Zebrafish models of myeloid disorders, this part also requires the schematic representation of its various functions.

6.2., 6.3., 6.4., This part requires more elaboration with relevant references.

There should be one more heading as future prospects.

7. Conclusions

This part needs very much refinement and includes all contents summary.

Overall comments

This is a review article which needs a lot of improvement of language and rephrases. This topic is relevant topic and authors tried to put many information in a single platform. The article nddes a major revision and authors are encouraged to resubmit the article. 

Author Response

Zebrafish as a genetic model for primary immunodeficiency (PID) disorders

Title: The present title seems very rudimentary so needs to put some more effort to revise the title and it should end with question mark. Change the title to make it more catch

  • As suggested, the title has been changed, including ending with a question mark, to make it more provocative.

Abstract:

In abstract section a flow is missing so needs to be rephrased. Need to mention the significance of the Zebra fish. This part is lacking of objective and scope of the MS. What it will add to existing knowledge.

  • The text has been rewritten to address these suggestions.

Introduction

 I suggest authors to put 80 % references should be beyond 2018. The old references can be replaced with latest references.

  • Considerable efforts have been made to include more recent references. However, there many seminal studies earlier than 2018 for which there is no alternative reference available. That said the proportion of 2018+ references has gone from <40% to nearly 60%, with another 20%+ in the preceding period 2014-7.

Author are suggested to following reference as standard reference.

 DK Meena, BK Behera, Das Pronob, AK Prusty, Kumar Satendra, M Sekar, Meena Kanti (2012). Transposable elements: strategies and mechanism of transposition in Danio rerio, a genetic model. Asian Journal of Bio Science.7(2):223-229.

  • Reference format changed to match that in the IJMS template provided.

Need to extend the introduction part.

  • As suggested, the introduction has been expanded, including a new figure (Figure 1).
  1. The zebrafish immune system, this heading can be changed as, The zebrafish immune system v/s immune system of higher vertebrates.
  • The heading has been modified such that the suggestion has been captured.
  1. Zebrafish models of lymphoid disorders, this part can be represented a schematic diagramme of various.
  • As suggested, a schematic of lymphoid disorders has been included (new Figure 1, panel A).
  1. Zebrafish models of myeloid disorders, this part also requires the schematic representation of its various functions.
  • As suggested, a schematic of myeloid disorders has been included (new Figure 1, panel B).

6.2., 6.3., 6.4., This part requires more elaboration with relevant references.

  • As suggested, these sub-sections have been expanded.

There should be one more heading as future prospects.

  • Future prospects have been included as part of the new ‘Limitations and Future Prospects’ Section.
  1. Conclusions

This part needs very much refinement and includes all contents summary.

  • As suggested, the Conclusion has been modified to better reflect a summary of the contents of the review.

Overall comments 

This is a review article which needs a lot of improvement of language and rephrases. This topic is relevant topic and authors tried to put many information in a single platform. The article nddes a major revision and authors are encouraged to resubmit the article. 

  • We thank the reviewer for the encouragement to re-submit.

Round 2

Reviewer 2 Report

The requested changes were made to this manuscript and I recommend that it be published.

Author Response

We thank this reviewer for their kind recommendation to publish the revised review manuscript.

Reviewer 3 Report

Authors tried to comply the comments however, i have suggested to add one relevent references that has not been included.

 DK Meena, BK Behera, Das Pronob, AK Prusty, Kumar Satendra, M Sekar, Meena Kanti (2012). Transposable elements: strategies and mechanism of transposition in Danio rerio, a genetic model. Asian Journal of Bio Science.7(2):223-229. 

Author Response

Authors tried to comply the comments however, i have suggested to add one relevent references that has not been included.

 DK Meena, BK Behera, Das Pronob, AK Prusty, Kumar Satendra, M Sekar, Meena Kanti (2012). Transposable elements: strategies and mechanism of transposition in Danio rerio, a genetic model. Asian Journal of Bio Science.7(2):223-229.

  • We thank the Reviewer for the suggested reference on the topic of transposable elements in zebrafish. However, transposable elements have not been utilized at all to my knowledge in the development or use of zebrafish models of human PIDs or their various application. Therefore it is difficult to see how the suggested reference can be reasonably incorporated in a meaningful way into our manuscript and so we have not included it.